

# Estimating the potential biodiversity impact of redeveloping small urban spaces: the Natural History Museum's grounds

Helen R.P. Phillips[1,2,3,4], Sandra Knapp[2] and Andy Purvis[1,2]

[1] Department of Life Sciences, Imperial College London, London, United Kingdom
[2] Department of Life Sciences, Natural History Museum, London, London, United Kingdom
[3] German Centre for Integrative Biodiversity Research (iDiv) Halle-Jena-Leipzig, Leipzig, Germany
[4] Leipzig Universität, Leipzig, Germany

Corresponding author
Helen R.P. Phillips,
helen.phillips@idiv.de,
helen.phillips11@imperial.ac.uk

## ABSTRACT

**Background**. With the increase in human population, and the growing realisation of the importance of urban biodiversity for human wellbeing, the ability to predict biodiversity loss or gain as a result of land use change within urban settings is important. Most models that link biodiversity and land use are at too coarse a scale for informing decisions, especially those related to planning applications. Using the grounds of the Natural History Museum, London, we show how methods used in global models can be applied to smaller spatial scales to inform urban planning.

**Methods**. Data were extracted from relevant primary literature where species richness had been recorded in more than one habitat type within an urban setting. As within-sample species richness will increase with habitat area, species richness estimates were also converted to species density using theory based on the species–area relationship. Mixed-effects models were used to model the impact on species richness and species density of different habitat types, and to estimate these metrics in the current grounds and under proposed plans for redevelopment. We compared effects of three assumptions on how within-sample diversity scales with habitat area as a sensitivity analysis. A pre-existing database recording plants within the grounds was also used to estimate changes in species composition across different habitats.

**Results**. Analysis estimated that the proposed plans would result in an increase of average biodiversity of between 11.2% (when species density was modelled) and 14.1% (when within-sample species richness was modelled). Plant community composition was relatively similar between the habitats currently within the grounds.

**Discussion**. The proposed plans for change in the NHM grounds are estimated to result in a net gain in average biodiversity, through increased number and extent of high-diversity habitats. In future, our method could be improved by incorporating purposefully collected ecological survey data (if resources permit) and by expanding the data sufficiently to allow modelling of the temporal dynamics of biodiversity change after habitat disturbance and creation. Even in its current form, the method produces transparent quantitative estimates, grounded in ecological data and theory, which can be used to inform relatively small scale planning decisions.

## INTRODUCTION

Urbanisation has increased globally and will continue to do so (*Heilig, 2012*). Urban expansion has resulted in the widespread loss, both directly and indirectly, of natural and semi-natural habitats which are important as refuges and corridors for biodiversity (*Goulson et al., 2002*; *Osborne et al., 2008*) and for human well-being (*Fuller et al., 2007*; *Dallimer et al., 2012*; *Bratman et al., 2015*; *Shanahan et al., 2016*). Any retention or creation of green-spaces within urban areas is therefore considered important (*Alvey, 2006*, but see *Deaborn & Kark, 2010*).

Urban ecology has become increasingly popular over the last decade (*McPhearson et al., 2016*). Several UK-based projects have assessed urban biodiversity (*Gaston et al., 2004*; *Angold et al., 2006*), investigating how it can be maintained and improved. Communicating the benefits of urban gardens and public spaces can result in enhanced biodiversity potential (e.g., *Thompson, 2007*).

The Natural History Museum, London, is a popular attraction, with around five million visitors per year—a number that is expected to increase. In part to alleviate the pressure of such large visitor numbers on the two current entrances, a third entrance through the Darwin Centre, at the west of the building, has been proposed. In order to comply with local council requirements for a unified theme between the museum building and the grounds (See Text S1 for further information), the proposed plans contained an overarching continuous theme similar to that of the museum building itself, moving from "extinct" habitats in the east to current British habitats in the west (Fig. 1B). The plans as proposed will result in the loss and reduction of some habitats within the grounds, gain and expansion of others, and disturbance particularly in the eastern part of the grounds. The proposed changes prompted concerns for the wildlife currently inhabiting the grounds, especially in the Wildlife Garden (henceforth WLG) in the southwestern corner of the site (*Knapton, 2015*; *Prospect, 2015*; *Doward, 2015*; *Duell, 2015*; *Marren, 2015*): a petition to stop the redevelopment of the grounds attracted over 37,000 signatures as of 1 June 2016 (*Weiler, 2015*).

Among the arguments used by critics of the proposals is that the grounds harbour unusually high levels of biodiversity (*Weiler, 2015*), which would be jeopardised by the proposed changes to the grounds (*Marren, 2015*). However, few quantitative tools exist to assess the levels of biodiversity within the grounds currently, and how it might change as a result of the redevelopment. Over 2,800 species have been recorded from the WLG in the 21 years since its creation, in occasional structured surveys and more haphazard observations (*Ware et al., 2016*). Despite species having been recorded since 1995 when the WLG was created, new species continue to be added to the cumulative list of taxa recorded (see Text S1 for additional information). However, lengths of lists of recorded species can
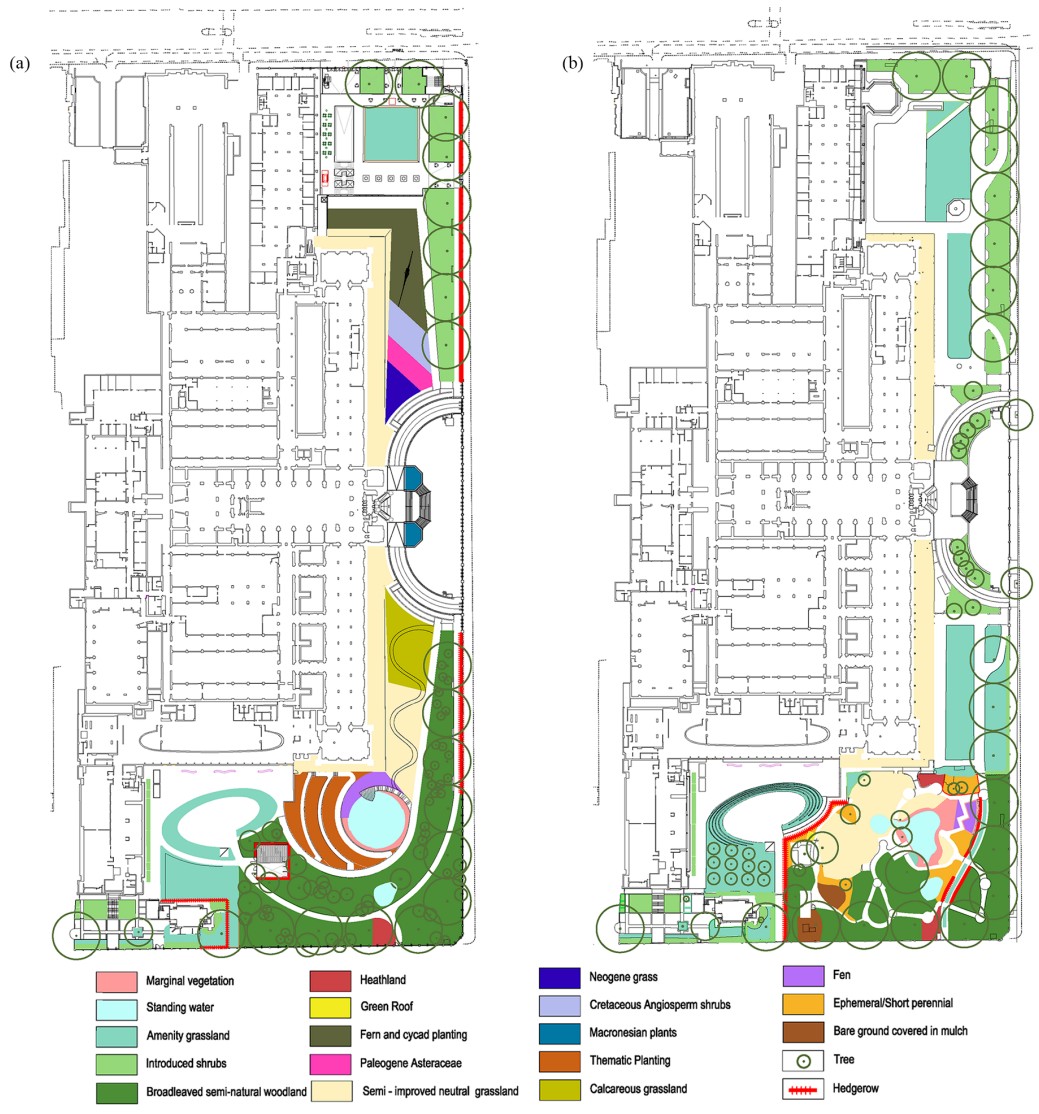

**Figure 1** **Detailed plans of the grounds of the Natural History Museum.** Detailed plans of the NHM grounds, provided by Wilder Associates, and the area (m$^2$) of: (A) Current habitat types and, (B) Proposed habitat types.

only be compared meaningfully across sites if sampling effort has itself been recorded or, better yet, been equal at each site; otherwise, lengths of lists typically conflate differences in sampling effort with true diversity differences (*Gotelli & Colwell, 2001*; *Crawley, 2005*). For example, assiduous sampling led to 2,204 species of plant and animals from selected groups being recorded over 15 years from a domestic garden in Leicester (*Owen, 1991*). Because species in many high-diversity taxonomic groups, such as insects or other invertebrates, can often be differentiated only by taxon specialists, taxonomic expertise can also influence lengths of species lists (*Crawley, 2005*). The Natural History Museum provides one of the greatest concentrations of such expertise in the world, meaning the list of species from the

WLG is likely to be more comprehensive than those from almost anywhere else on earth. In addition, it is not possible to determine how lengths of species lists will change over time.

The most accurate prediction possible of the effects of a redevelopment such as this would be obtained by extensive standardised ecological sampling of the site to provide a precise estimate of current biodiversity, together with similar sampling of nearby patches of any habitat types that would be added as a result of the redevelopment. However, planning decisions are usually based on much less detailed information than this.

An independent ecological assessment of the biodiversity value of the WLG, done as part of a planning application, suggested that, apart from breeding birds, a number of invertebrates and the accidental introduction of a slow worm, "No other protected or noteworthy species were considered likely to be supported within the site" (*The Ecology Consultancy, 2015a*; *The Ecology Consultancy, 2015b*), although some protected species (common and soprano pipistrelle bats) had been seen foraging in the garden. These findings are in line with expectations for young anthropogenic habitat patches in an urban setting. However, as with the lists of recorded species, this assessment did not provide any quantitative estimates of diversity that could provide the basis of a comparison between the biodiversity of the current grounds and that expected or (in future) found under the new proposal.

Few robust tools are available to estimate potential impacts to biodiversity from development and land-use change, especially at such small spatial scales. For planning applications it is advised, although not always a necessity, that ecological surveys (desk-based or field-based surveys as part of a Preliminary Ecological Assessment and/or an Ecological Impact Assessment) be conducted prior to submission to determine, amongst other things, how species and habitats at the site might be impacted by the proposed works (*CIEEM, 2016*). However, especially with desk-based surveys, these methods would be unable to estimate the likely gains or losses of biodiversity until after the fact. DEFRA's Biodiversity Offsetting model (*DEFRA, 2012*) offers a potential way of assessing potential impact on biodiversity via the habitat types that are to be displaced: briefly, each habitat type carries a distinctiveness score (2, 4 or 6), each patch is assigned a condition score (1, 2 or 3), and these are multiplied together to calculate a per-hectare biodiversity score which is multiplied by the area of the habitat patch and summed across all patches to give an overall biodiversity score. For increased or new areas of biodiversity-rich habitat, scores are moderated to reflect the time needed to achieve the target level of biodiversity and the risk that it will never be reached. In order to prevent net loss of biodiversity, the score of the proposed habitat types would need to match or exceed the score of the habitats being displaced. Although operational, this offsetting method falls short in urban environments (habitats are presumed to be in a natural setting), and the scores are not strongly grounded in relevant biodiversity data (see *Baker et al., 2014*).

A common approach in conservation ecology to the problem of estimating the effects of land-use change on biodiversity is to undertake comparable ecological surveys at nearby sites in different land uses, under the assumption that such spatial comparisons can be used in lieu of time-series data tracking biodiversity through land-use changes. Although no such data have been published from within the WLG itself, such comparisons

are sufficiently common to permit powerful global syntheses (e.g., *Alkemade et al., 2009*; *Gibson et al., 2011*; *Gerstner et al., 2014*). In particular, the PREDICTS project has modelled data from surveys worldwide to estimate how land-use change and related pressures affect occurrences and abundances of many species (*Newbold et al., 2014*; *De Palma et al., 2015*) and broader site-level measures of biodiversity (*Newbold et al., 2015*; *Newbold et al., 2016*; *De Palma et al., 2016*). By focusing on surveys that have included sites in different land uses, this approach is able to estimate relative levels of biodiversity for each land use type, even if no single survey represents the full range of land uses. By empirically describing the relationship between pressure data and the response of biodiversity, (i.e., using a dose–response modelling framework: *Pereira et al., 2010*), the model can be combined with projections of future pressures (e.g., land use) to estimate average levels of site-level biodiversity in the future, enabling comparison with the present (*Newbold et al., 2015*). The PREDICTS framework is therefore designed to tackle similar kinds of question to those posed by the museum's grounds redevelopment, such as, will the development cause a negative effect on biodiversity over the long term?

Given this conceptual similarity, aware of the controversy surrounding the biodiversity costs and benefits of the proposed development, and having no involvement in either the conception of the proposal or the opposition to it, two of us (HRPP and AP) offered to undertake an analysis for the Natural History Museum, conceptually derived from that of *Newbold et al. (2015)*, to estimate the net effects of the proposal on biodiversity and to make the resulting estimate public. The proposal was accepted by the Natural History Museum, on a short three-month timescale. SK, already involved in the Grounds Transformation Project, joined the analysis and provided detailed information about the current and proposed layouts of the grounds, as well as facilitating access to the dataset of species recorded from the various habitats within the WLG.

Our aim was to provide quantitative estimates of biodiversity metrics in the current NHM grounds and the corresponding values following the redevelopment. In estimating the biodiversity consequences of the proposed redevelopment, we are also aiming to develop a decision-support approach that, while undoubtedly less accurate than extensive bespoke ecological surveys, provides estimates that are quantitative, transparent and data-based in a reasonable time and at a reasonable cost. We extended the analytical framework developed by *Newbold et al. (2015)* to allow for the fact that the spatial extent of a habitat, as well as its type, is likely to affect its biodiversity value. Larger habitat patches are expected to contain not only more species overall than smaller patches (in line with the species–area relationship: e.g., *Rosenzweig, 1995*; *McKinney, 2008*), but also—though less strongly—more species per unit area (i.e., the species density-area relationship: *Phillips et al., in press*). Our analysis aims to take such area-dependency into account. Many other factors can also affect site-level diversity, notably habitat age (*Sattler et al., 2010*), edge effects (*Murcia, 1995*), vegetation structure (*Threlfall et al., 2017*), and habitat connectivity (*Shanahan et al., 2011*). We return to these in the discussion, but time constraints (in order to report in time to feed into the planning application) and limited data availability precluded their consideration in this study.
## METHODS

### Study site

The Natural History Museum in London (NHM) has 2.18 hectares of grounds around the buildings at its main South Kensington site, which for ease of reference can be split (at the centre of the museum building) into the "eastern area" and "western area" (Fig. 1A). The museum grounds were renovated in 1995 with the creation of a one-acre (0.4 hectares) Wildlife Garden (WLG; *Honey, Leigh & Brooks, 1999*) in the western area, which contains small areas of multiple lowland habitats present in southern England. The eastern area is heavily and repeatedly disturbed due to temporary attractions (a butterfly exhibit in the summer and ice rink in the winter); at other times, it contains only regularly-replaced amenity grassland and areas of introduced shrubs with no habitats traditionally considered "wildlife-friendly".

The entire green space comprising the grounds has been designated a non-statutory Site of Borough Importance for Nature Conservation (SINC) grade II, and is in close proximity to two other non-statutory SINCs; (i) Prince's Gate East, Prince's Gate West and Rutland Gate North, and (ii) Hyde Park and Kensington Gardens. The NHM grounds, both current and post-renovation, were classified into 19 different habitat types, terrestrial and aquatic, some of which can be linked to the UK BAP Broad habitat classes (Table 1).

### Biodiversity measure

Biodiversity is a complex, multifaceted and multiscale concept that cannot be captured fully by any single measure (*Purvis & Hector, 2000*). Given time constraints, we therefore had to choose the most appropriate measure of biodiversity to include in our models. Perhaps the most intuitively appealing would be the overall species richness of the grounds. However, as outlined above, the sampling undertaken so far does not provide a basis for estimating this quantity in the present, and even if it did there would be no basis for estimating overall species richness under the proposed changes.

*Newbold et al. (2015)* focused mainly on within-sample species richness and overall abundance, both expressed relative to the values expected for a pristine site (i.e., a site with no human impacts). Such a baseline is not appropriate for young anthropogenic urban habitats, which are typically not expected to approach the diversity of pristine habitats and which are not in close geographic proximity to any such habitats. Additionally, *Newbold et al. (2015)* did not consider the effects of habitat patch size on within-sample species richness, despite the expectation of a positive correlation (*Phillips et al., in press*). To overcome these twin limitations, we chose to use a measure of biodiversity that can incorporate effects of patch size—namely species density (the expected number of species sampled in a constant area of a given habitat; *Whittaker, Willis & Field, 2001*; *Magurran, 2004*)—and did not attempt to express values relative to a pristine baseline.

### Collation of data

We conducted literature searches to identify publications that compared within-sample species richness between two or more of the habitat types in Table 1. Two searches were undertaken: the first set of search terms was highly specific (full search terms in Text S2)

Phillips et al. (2017), *PeerJ*, DOI 10.7717/peerj.3914

**Table 1** Habitat types in the current and the proposed plans of the Natural History Museum grounds.

| Habitat type | Description | UK BAP broad habitat | Current area (m$^2$) | Proposed area (m$^2$) | Coefficient |
|---|---|---|---|---|---|
| Hard-standing | Pathways and other concreted areas | NA | 10,415 | 9,525.16 | Assumed to be zero |
| Amenity grass/turf | Gardens, lawns or turfed areas | NA | 3,303.63 | 1,573.91 | Modelled |
| Introduced shrubs | Beds planted with introduced species, with occasional trees | NA | 2,218.62 | 1,346.69 | Broadleaved woodland coefficient adjusted based on *Strong & Levin (1979)* |
| Neutral grassland | Rotational grazing by sheep during late summer months and autumn. Area estimates include the semi-improved grassland | Neutral grassland | 2,103.15 | 2,133.45 | Modelled |
| Broadleaved woodland | Mixed tree species, usually dominated by pedunculate oak (Quercus robur) and silver birch (Betula pendula), understory typically comprised of hazel (Corylus avellana) and holly (Ilex aquifolium) | Broadleaved, mixed and yew woodland | 1,978.36 | 3,477.67 | Modelled |
| Short/perennial vegetation | Ephemeral vegetation, such as common nettle (Urtica dioica), dandelion (Taraxacum officinale agg.) and creeping buttercup (Ranunculus repens) | NA | 423.65 | 0 | Modelled |
| Chalk grassland | Species richness grassland, abundant species include kidney vetch (Anthyllis vulneraria) and sheep's fescue (Festuca ovina) | Calcereous grassland | 344.58 | 526 | Modelled |
| Ponds | Currently three ponds (70 m$^2$, 90 m$^2$ and 400 m$^2$) with linked water systems. Designed to be typical of chalk and peat ponds, but currently contain similar plant communities. Proposed plans contain two ponds | Standing water and canals | 341.28 | 459.37 | Modelled |
| Marginal vegetation (pond edge) | Pond surrounding, dominated by common reed (Phragmites australis) | Standing water and canals | 163.6 | 99.15 | Modelled |

Peer**J**

**Table 1** (*continued*)

| Habitat type | Description | UK BAP broad habitat | Current area (m²) | Proposed area (m²) | Coefficient |
|---|---|---|---|---|---|
| Species-rich hedgerow | Hedgerow with more than one native species, typically dominated by hawthorn (Crataegus monogyna) | Boundary and linear features | 121.87 | 607.5 | Modelled |
| Species-poor hedgerow | Single species hedgerow | Boundary and linear features | 109 | 0 | Species-rich hedgerow coefficient adjusted based on *Scriven, Sweet & Port (2013)* |
| Acid grassland (heath) | Included both wet and dry acid grassland | Dwarf shrub heath | 100 | 82 | Modelled |
| Fen (including reedbed) | Fen species included marsh fern (Thelypteris palustris), common reed (Phragmites australis) and lesser pond sedge (Carex acutiformis) | Fen marsh and swamp | 64.6 | 133.86 | Modelled |
| Green roof | Planting on top of shed | NA | 9.98 | 0 | Modelled, based on Canadian study |
| Ferns and cycad planting | Plantings of (predominantly) non-native ferns and cycads | NA | 0 | 729.82 | Introduced shrubs coefficient |
| Agricultural plants | Rotating crop plantings, species similar to those planted in allotments | NA | 0 | 583.97 | Modelled |
| Cretaceous Angiosperm shrubs | Angiosperms similar to those present during the late Cretaceous period | NA | 0 | 244.97 | Broadleaved woodland coefficient adjusted based on *Strong & Levin (1979)* |
| Paleogene Asteraceae | Asteraceae similar to those present during the Paleogene period | NA | 0 | 176.57 | Short/perennial vegetation coefficient |
| Neogene grass | Grass similar to that present during the Neogene period | NA | 0 | 156.27 | Amenity grass/turf coefficient |

**Notes.**
For each habitat type a brief description is given, its UK BAP Broad Habitat classification, current area and area under the proposed plans and how the coefficient for the biodiversity estimate was obtained.

while the second—to fill the many remaining gaps—was broader (full search terms in Text S3). Additional searches targeted habitats for which data were lacking, particularly habitats which are not typically urban or widespread in the UK.

We used data collected from urban environments wherever possible (three published articles were included from non-urban environments, as these provided data from habitats not typically found in urban habitats, or provided comparisons of habitats where data was lacking: *Petit & Usher, 1998*; *Wilson et al., 2003*; *Williams, Whitfield & Biggs, 2008*). Data had to meet four criteria:

1. The study sampled invertebrates and/or plants in more than one habitat type and/or within a habitat of differing area or age.
2. Sampling was undertaken within the UK (with the exception of samples from one published article on green roofs, as no suitable UK data were found).
3. The paper presented the area over which the sampling was conducted; this area was either the sampling frame or the size of the patch of habitat (if the entire patch was sampled).
4. Data were presented as species richness values, although abundance measures were also recorded if presented.

ImageJ (*Schindelin et al., 2012*) was used to extract data from figures when species richness values were not provided in text form. We did not find sufficient data that compared habitats of different ages or that reported measures of abundance, so these aspects of the original design of the study were dropped for practical reasons in order to meet the planning deadline.

The data from each paper were collated as a "study". If a paper contained data from multiple sampling methodologies then it was split into multiple studies based on the methodology (following *Hudson et al., 2014*). Data were recorded for each site within a study where possible, or otherwise as averages/totals for each habitat type within a study. For each study, we recorded whether it sampled invertebrates or plants. We classified the habitat of each site into one of the 19 habitat types in Table 1; any sampled habitats not present in the museum's grounds or renovation plans were excluded from the analysis. (All data extracted from the literature is available at http://data.nhm.ac.uk/dataset/grounds-metaanalysis-data and code is available at https://github.com/helenphillips/GroundsRenovation).

## WLG plant database

Data on plants from the WLG database were also included in the modelling dataset to increase the robustness of some habitat comparisons. The WLG is currently split into 55 zones of different size (see Fig. 1 in *Leigh & Ware, 2003*), with each zone's assemblage originally planted based on National Vegetation Classification communities (*Rodwell, 1998*; *Honey, Leigh & Brooks, 1999*). Between 1995 and 2015 a complete inventory of the plant species in each zone has been completed non-systematically every year. Because the database species binomials included some synonyms, species names of all records were standardised using the UK Species Inventory (UKSI) database (*Raper, 2014*). Current WLG habitat types of each zone were taken from (*Leigh & Ware, 2003*) and confirmed by WLG habitat managers. With advice from members of the Grounds Project team, we classified

each zone into a habitat type (Table 1) and the species richness of each zone was calculated as the total number of species surveyed between 2013 and 2015 (on the grounds that species might be missed in any year, and that more recent surveys are more relevant to the current state). Each zone was treated as a site, with the area estimated through digitisation of Fig. 1 in *Leigh & Ware (2003)*. Although the WLG database also contains data on other groups of organisms, such as invertebrates, these were not suitable for our analysis as sampling effort and methodology were too heterogeneous.

## Accounting for area effects

As well as depending on the nature of the habitat, the expected number of species in a sample also depends on the area covered by the sample (the species–area relationship, or SAR: *Rosenzweig, 1995*) and the extent of the (often much larger) habitat patch within which the sample was taken (the species density-area relationship, or SDAR: *Phillips et al., in press*).

Samples covering larger areas will encompass a wider range of microclimatic and other environmental conditions, meaning that more species have the potential to be sampled. Larger patches of habitat can additionally support larger populations of resident species meaning that species density is likely to be higher. Both of these relationships need to be considered in order to provide the best estimate of the net effects of the proposed redevelopment on biodiversity within the grounds, especially if there is a mismatch in habitat areas being predicted and the areas from which sampled diversity estimates are taken.

We estimated the expected species density for a 10 m$^2$ sampling frame, from each site's within-sample species richness and area sampled, using:

$$\log S_{10} = \log S_s + z(\log 10 - \log A_s)$$

Where $A_s$ is the area over which the sample was taken and $S_s$ the number of species in the sample, and 10 is the area for which species density ($S_{10}$) is calculated for. Theory predicts that $z \sim 0.10$ (*Phillips et al., in press*): the difference between the island SAR for isolated fragments ($z \sim 0.25$) and the continental SAR ($z \sim 0.15$). *Phillips et al. (in press)* tested this prediction, estimating z empirically from a synthesis of data from 38 studies; the empirical estimate was $z = 0.07$, but the predicted value fell within the 95% confidence interval (0.048 to 0.11). We therefore use $z = 0.10$ in the analyses that follow, but present results of $z = 0.7$ in Text S4 as a further sensitivity analysis. Because the area-scaling of species density is not yet well established (e.g., *Giladi et al., 2014*), we also modelled within-sample species richness as a response variable.

## Modelling

A generalised linear mixed-effects model (*Bates et al., 2015*) was used to estimate average species density (per 10 m$^2$) and species richness for each habitat type. Both response variables were rounded to the nearest integer to allow for the appropriate error structure, as count data are expected to follow a Poisson distribution. Study identity was included as an intercept-only random effect to account for differences in methodology and the resulting heterogeneity of the data (*Zuur, Ieno & Saveliev, 2009*). The maximal models

included habitat type with an additive effect of taxonomic group as fixed effects; there was not enough data to create a meaningful interaction between the two main effects. Model simplification was based on log-likelihood ratios (*Zuur, Ieno & Saveliev, 2009*; *Crawley, 2012*); main effects were removed if $P > 0.01$.

Six habitat types (Table 1) were not represented by enough data for an average species density or species richness to be modelled. For these six habitats, species diversity (density and richness) was estimated either using the modelled coefficient from another, similar, habitat type; or by using a single study to relate species diversity to the estimated coefficient for another habitat type. The last column of Table 1 gives details of these estimates. Additionally, we assumed that hard standing (asphalt and pavement) had zero species richness.

## From statistical models to estimates of biodiversity

For both the current grounds and the proposed redevelopment, we combined the areas of each habitat type with the coefficients of our models in order to estimate overall biodiversity, so that these estimates could be compared to assess the net changes. We explored the effects of three alternative assumptions when using our model coefficients.

**Assumption 1** (**Area-scaling of both input data and model output**): For each habitat patch, we used the appropriate coefficient from our model of species density, but rescaled it to the area of the habitat patch to reflect the area-scaling of species density. Scaling species density for habitat area assumes that the habitat is effectively contiguous (i.e., any breaks in the habitat do not prevent movement or dispersal across them). Although this is typically the case in the renovation plans, it is less so in the current grounds. Thus, any bias caused by this assumption will tend to overestimate the overall biodiversity value of the current grounds.

**Assumption 2** (**Area-scaling of input data only**): For each habitat patch, we used the appropriate coefficient from our model of species density, but did not rescale it to the area of the habitat patch.

**Assumption 3** (**No area-scaling**): For each habitat patch, we used the appropriate coefficient from our model of within-sample species richness. Most comparisons of species richness among habitats do not consider effects of area on the numbers of species sampled at all; we therefore also modelled this possibility.

For each assumption in turn, and for each layout (current or proposed), we computed the area-weighted sum of habitat scores; i.e., each habitat's biodiversity score was multiplied by its area in that layout, and the products summed across all habitat patches. Within each assumption, these scores can be compared between the current and proposed layouts.

## Sensitivity analysis

A sensitivity analysis was performed to assess the robustness of the modelled species density coefficients under the three assumptions. For each of the 19 habitats, a normal distribution was created where the mean was the estimate of species density (per $10\,\text{m}^2$) and the standard deviation the standard error (*Newbold et al., 2015*). For the six habitats without modelled

coefficients, the means were calculated as above (Table 1), with standard errors of the same habitat type also being used but multiplied by 1.5 to reflect the increased uncertainty. The total weighted species density values for before and after the grounds renovation were calculated, as above, and the percent change between the two recorded. This process was repeated 1,000 times and the frequency of negative change (i.e., biodiversity loss under the proposed plans) determined.

## Compositional similarity

Community similarity of the habitats within the current WLG was estimated using the plant database as an indication of how overall species composition might change with the removal of some habitats. Using just the records in the database between 2013 and 2015, the similarity of species composition (percentage of species in common) was calculated between each pair of habitat types, and the results displayed as an asymmetrical matrix. Thus, for each habitat on the $x$-axis, the matrix shows the percentage of species that habitat-x shares with a habitat on the $y$-axis.

# RESULTS

## Meta-analysis

The first literature search returned 101 articles and the second found 1,158 articles. Further targeted searches acquired data from an additional five articles. Based on the data criteria, only data presented in 11 papers were suitable for modelling; these were collated into 14 studies based on methodology. These studies contained sampled sites from across the UK (Fig. 2), as well as two studies in Canada (to allow a comparison to green roofs: *MacIvor & Lundholm, 2011*), and included suitable data we were able to access from the WLG database.

The fixed effects of the mixed-effects model of species density (per 10 m$^2$) were simplified by the removal of the additive effect of taxon ($\chi^2 = 0.90$, *d.f.* = 1, *p*-value = 0.34). Species density (per 10 m$^2$) varied significantly between habitat types ($\chi^2 = 353.18$, *d.f.* = 12, $p < 0.01$; Fig. 3). Chalk grassland had the highest species density (per 10 m$^2$), whilst pond and fen had the lowest among habitats for which sample-based data were available.

Similar to the species density model, the model of within-sample species richness was also simplified with the removal of the additive effect of taxon ($\chi^2 = 3.29$, *d.f.* = 1, *p*-value = 0.07). Species richness significantly varied among habitat types ($\chi^2 = 468.01$, *d.f.* = 12, $p < 0.01$; Fig. 3). The relative diversity of each of the habitats was largely consistent among the two models.

Calculations for all three assumptions indicate an overall net increase in local biodiversity with the proposed plans for the museum's grounds. Assumption 1 yields an increase of 11.17%. Under Assumption 2, the increase is estimated to be 13.20%. Assumption 3 gave the greatest increase (14.05%) in overall net biodiversity under the proposed plans.

## Sensitivity analysis

When the analysis was repeated 1,000 times, taking the habitat coefficients from a distribution, the proposed plans only resulted in a net loss of biodiversity in 0.4% of the trials under Assumption 1 (Fig. 4) and never did so under Assumptions 2 and 3 (Fig. S5).
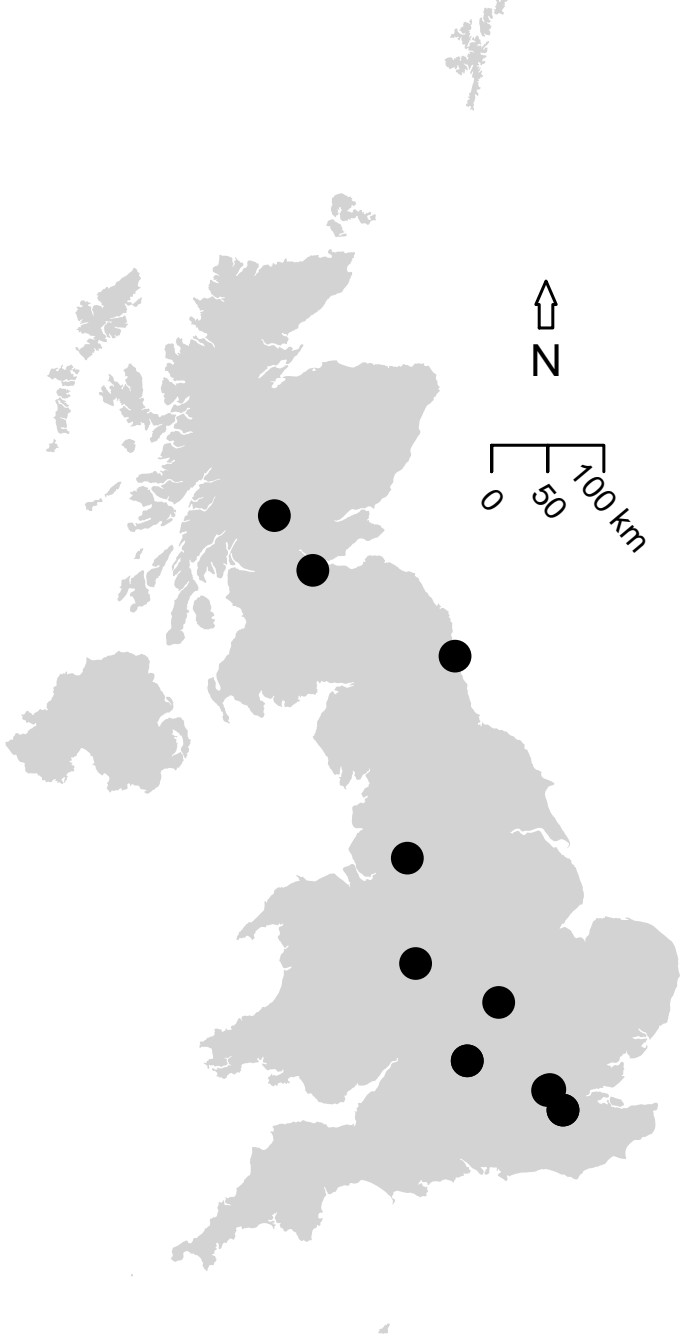

**Figure 2** **Map of the 12 UK studies (10 papers) included in the analysis.** Data Sources for this analysis. (*Petit & Usher, 1998*; *Wilson et al., 2003*; *Fountain & Hopkin, 2004*; *Smith, Chapman & Eggleton, 2006*; *Butt et al., 2008*; *Williams, Whitfield & Biggs, 2008*; *Scriven, Sweet & Port, 2013*; *Sirohi et al., 2015*; *Speak, Mizgajski & Borysiak, 2015*, WLG Database). *MacIvor & Lundholm (2011)* (a Canadian article on green roofs) was included in the analysis (containing two studies) but is not shown on this map. Size of the points do not indicate the study area or sample size. All data and code is available for download from: https://github.com/helenphillips/GroundsRenovation and http://data.nhm.ac.uk/dataset/grounds-metaanalysis-data.

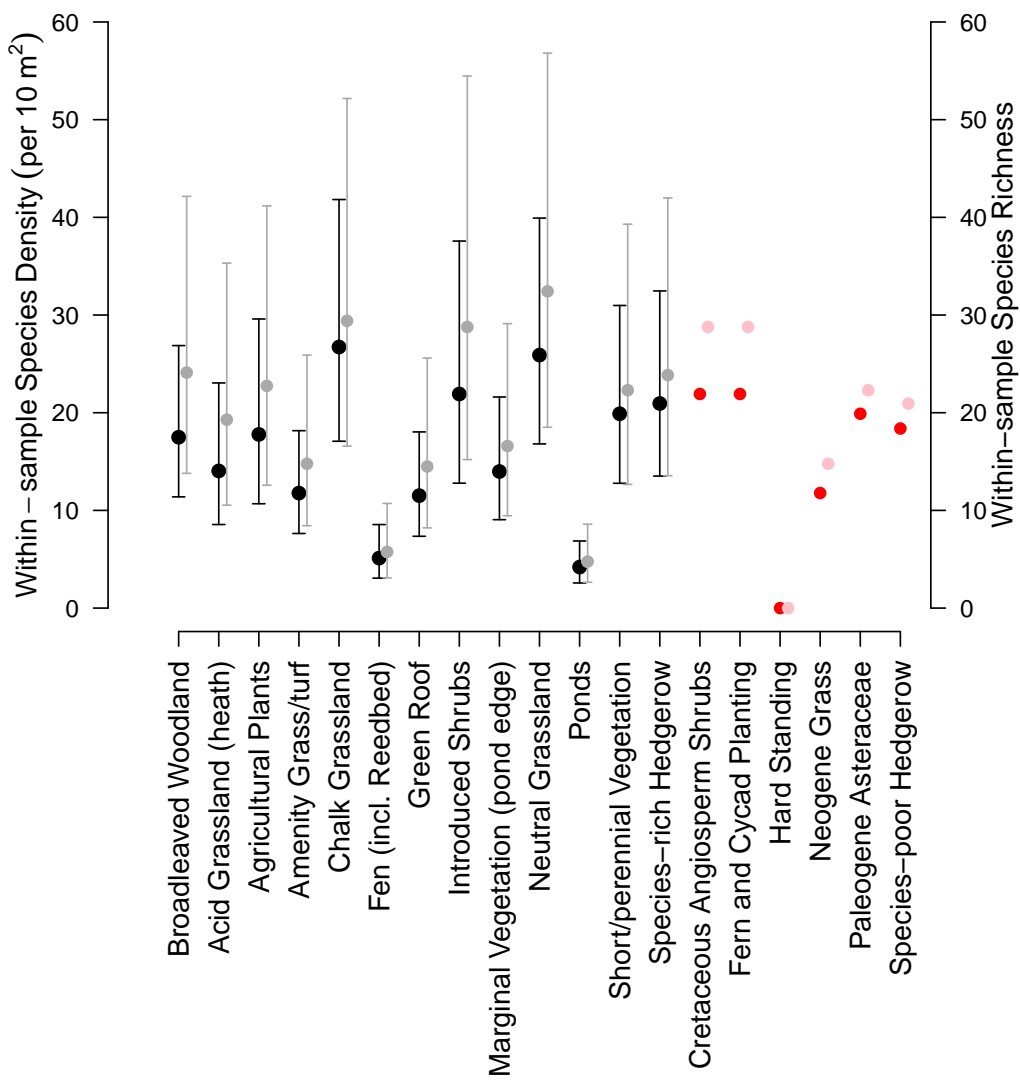

**Figure 3** **Model estimates of the 19 habitats within the Museum grounds.** Black coefficients are modelled species densities (10 m²), whilst red coefficients are the habitat densities that were unable to be modelled and estimated from other habitats (details in Table 1). Grey coefficients are modelled within-sample species richness and pink coefficients are the within-sample habitat richness of those unable to be estimated. Error bars indicate 95% confidence intervals.

## WLG species similarity

Most habitats had very similar plant species composition (Fig. 5), though there were exceptions. For example, very few species from other habitats were found in amenity grass/turf but nearly all species in amenity grass/turf were in most other habitats. Unsurprisingly, ponds had a highly dissimilar collection of species to every other habitat.

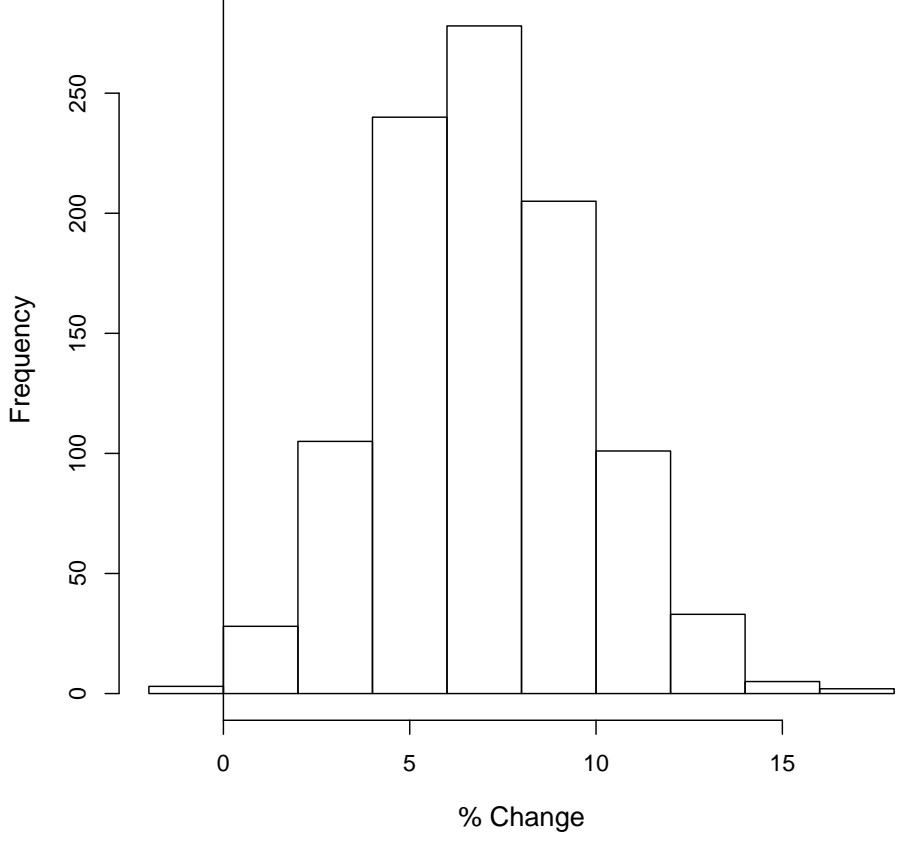

**Figure 4** **The number of times each percentage change in average species density was obtained in the sensitivity analysis.** A random sample was taken from the distribution of each habitat coefficient, and under Assumption 1 the overall gain or loss in average species density was calculated. This was repeated 1,000 times. Vertical line indicates 0% change. 0.4% of the runs resulted in a loss of species richness under Assumption 1.

## DISCUSSION

The findings of this meta-analysis indicate that the proposed plans for the museum grounds are expected to result in a net gain of local biodiversity. This increase in biodiversity is beneficial, not only to ease concerns of those that suspect diversity to be lost with the proposed grounds renovations (*Weiler, 2015*; *Marren, 2015*), but also as even a small increase in species richness in urban greenspaces has the potential to increase human wellbeing (*Fuller et al., 2007*; *Shanahan et al., 2016*). This increase in biodiversity arises because habitats with the highest modelled species density, such as chalk and neutral grassland, will increase in area under the new plans; and because new habitats will be introduced. These findings are similar to those of earlier studies; for instance, a previous synthesis of findings from studies worldwide investigating biodiversity in urban parks found that increasing the habitat area and habitat diversity usually increased species richness (*Nielsen et al., 2014*). Both area and number of habitats are likely to be important

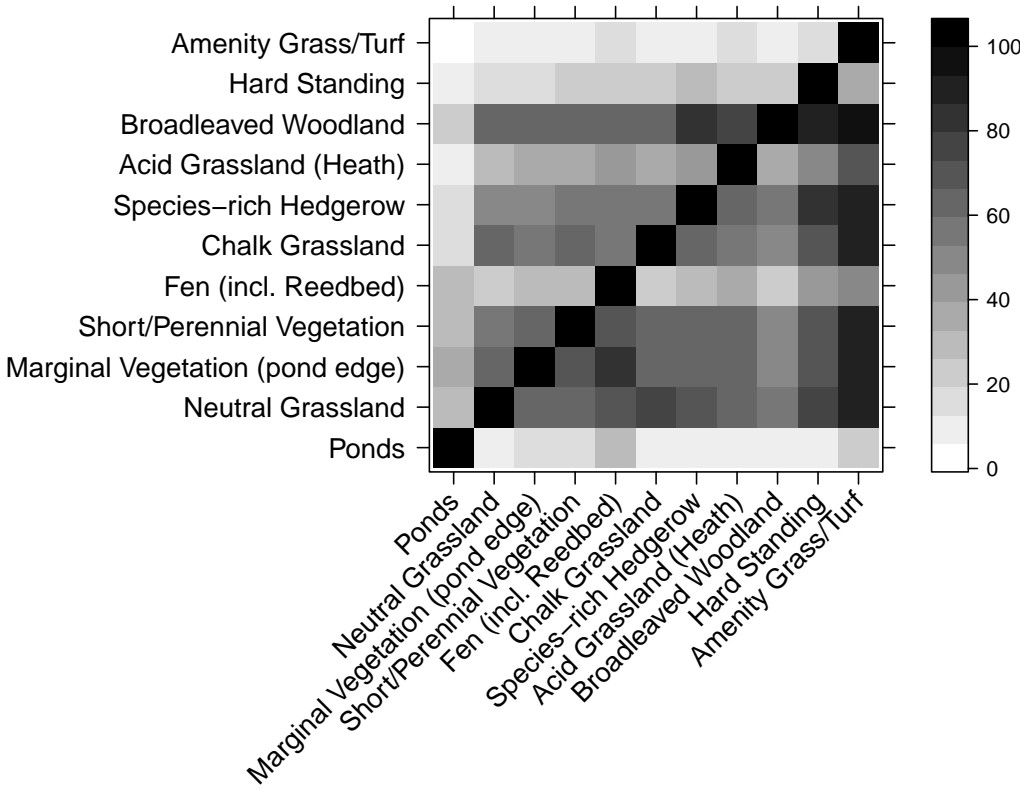

**Figure 5** **Compositional similarity between habitat types, based on data from the WLG database of plant species.** Each cell shows the percentage of species in the habitat on the *x*-axis that are also present in the habitat listed on the *y*-axis. Therefore, the grids above and below the diagonal are not mirror-images.

determinants of biodiversity, and potentially it would be more appropriate to incorporate them within a single model (e.g., the Choros model: *Triantis et al., 2003*).

Broadleaved woodlands and neutral grassland contribute greatly to the grounds' current biodiversity value, and will increase in extent under the proposed plans. As broadleaved woodland will be among the least disturbed habitats during the renovation process, this high-biodiversity-value area may harbour source populations for some of the other habitats, especially considering the relatively high proportion of shared plant species in all the other terrestrial habitats.

The statistical methods used in the analysis rely on species diversity modelled as comparisons between habitats that are present in the grounds currently or under the proposed plans. In this case it resulted in a relatively small sample size (14 studies from 11 papers), which was adequate for the model structure employed (i.e., models converged with acceptably narrow confidence intervals). This facilitates comparisons among the fully terrestrial habitats, where sampling methodologies are more likely to be consistent, but there are unsurprisingly few studies that use a consistent methodology between any fully terrestrial habitat and an aquatic habitat. Thus, the modelled coefficients for ponds are likely to be the least reliable in our analysis, as they are estimated from one comparison

against reeds (*Williams, Whitfield & Biggs, 2008*). However, as ponds make up only a small area of both the current and the proposed grounds (see Fig. 1), this will have little impact on the grounds' overall biodiversity value. Previous work has shown that diversity often increases with pond area (*Oertli et al., 2002*; *Parris, 2006*) and pond perimeter (*Gagné & Fahrig, 2007*; *Gagné & Fahrig, 2010*). With the ponds increasing in area under the proposed plans this may result in a relative increase in species richness compared to the current grounds. However, the proposed plans will reduce the extent of the pond's hard edges. Therefore, additional planting of marginal vegetation could increase the perimeter, whilst also increasing diversity (*Williams, Whitfield & Biggs, 2008*; *Gioria et al., 2010*).

The measures of biodiversity that we have modelled—species density and within-sample species richness—are pragmatic choices given the data available, but may not fully reflect desired features, such as the diversity of rare or charismatic species. Additionally, it is not possible to assess how communities within, and between, habitats might change (e.g., changes in the abundance distribution, or a shift to communities dominated by more widespread species). Capturing changes in beta diversity (i.e., spatial turnover) would allow the assessment of whether additional habitats are increasing the diversity of the grounds as a whole. Adding new habitats may not add species that are not already present in other habitats within the grounds, especially within an urban environment where habitats may not be as high-quality as in rural areas, (*Crooks, Suarez & Bolger, 2004*). Methods exist that analyse changes in community composition (such as Sørensen's similarity index; *Magurran, 2004*), which are much more sensitive than species richness to compositional change (*Hillebrand et al., 2017*); however, data extracted from papers suitable for this study often lacked diversity measures at the species level, therefore the use of such methods was not possible. In addition, we were unable to use rarefaction or abundance based metrics, as the numbers of individuals of each species were not known. However, with more detailed data, these metrics would provide an interesting avenue for future studies.

Other limitations and assumptions made in the analysis might also impact the results. In the calculations for the area-scaling of species density, habitat is assumed to be contiguous. Habitats are more contiguous in the proposed plans than in the current grounds, meaning that the proposed plans are disadvantaged in our comparisons. We focused on invertebrates and plants; although these are potentially the most appropriate taxa given the small size of the museum grounds, we would have liked to be able to also infer the response of vertebrates.

One of the main objections to the proposed renovation plans voiced by members of the public and other stakeholders is the level of disturbance that will be caused across much of the grounds and the potentially negative impact this will have on biodiversity. We had aimed to address this point by modelling how habitat age influences biodiversity but there were insufficient data for this analysis. In more natural settings, previous work has established that it can take many decades (*Hirst et al., 2005*)—even a century or more (*Vellend et al., 2006*)—for biodiversity to reach pre-disturbance levels. In urban settings, biodiversity is known to increase with habitat age (*Yamaguchi, 2004*; *Sattler et al., 2010*) and age of the surrounding city (*Aronson et al., 2014*). However, as diversity levels are typically lower than those of natural habitats (*Öckinger, Dannestam & Smith, 2009*; *Bates et al., 2011*), the time needed to recover could be considerably shorter. Considering that

the WLG is only 20 years old, it is unlikely the current biodiversity levels have reached equilibrium. Even if they have, the community composition is likely to be different from that of habitats in more natural settings (*Angold et al., 2006*).

Post-disturbance natural colonisation may be the main source for biodiversity recovery, and thus an important determinant of dynamics will be the connectivity of the museum grounds to potential source pools. Many studies have suggested that connectivity within an urban environment is important in maintaining biodiversity (*Öckinger, Dannestam & Smith, 2009*; *Goddard, Dougill & Benton, 2010*; *Kong et al., 2010*; *Vergnes, Le Viol & Clergeau, 2012*). However, an earlier study (*Angold et al., 2006*) reported that landscape variables, such as habitat connectivity, were less important than local site-level variables, such as site age or habitat size, for invertebrate communities in urban environments. This dichotomy of results could be due to the mobility of the studied taxa (*Braaker et al., 2014*), with highly mobile species benefiting from connectivity in the landscape more than less mobile species. Trait-based statistical models provide a possible approach to testing this possibility (e.g., *Öckinger et al., 2010*; *Lizée et al., 2011*).

Given the fragmented nature of urban landscapes, expectations of additional factors that could impact site-level diversity can be drawn from fragmented natural systems. In natural systems among the best-studied pressures associated with fragmentation are edge effects (disturbance from the surrounding matrix penetrating the habitat fragment; *Murcia, 1995*). Depending on the taxa studied, the impact from fragment edges can extend into the habitat fragment between 10 m and 2 km (*Broadbent et al., 2008*), potentially reducing the diversity (*Soga et al., 2013*) or increasing diversity with the movement of matrix species into the habitat fragment (*Ewers & Didham, 2006*). The lack of fragmentation data (e.g., data on the distance to the edge of the fragment) in the primary literature meant that we were unable to analyse this aspect. However, as the habitat areas within our studies from the primary literature were small (range = 7–6,250,000 m$^2$, median = 1,497 m$^2$; a range that largely overlaps the habitat areas in the current and proposed grounds), our sampled diversity estimates are already likely to have been impacted by edge effects (*Soga et al., 2013*). Therefore, although potential effects from habitat edges should ideally be tested and if appropriate incorporated into future modelling frameworks, in the context of this study the effects on the comparison between current and proposed grounds may be small.

Other factors that could impact site-level diversity relate to the landscape context, such as the similarity of the matrix habitat to that of the fragment (*Ruffell, Clout & Didham, 2017*) and the amount of similar habitat within the surrounding landscape (increasing the area from which colonists can arrive: *Shanahan et al., 2011*). The effects of these two factors, as well as the impact of edge effects, will differ between species. Meta-analytical studies, such as ours however, rely on the data presented within the original articles. In this study, none of the original studies provided quantitative data on these aspects of landscape context that could be meaningfully compared across different studies.

Although our results are unlikely to be directly transferable to other case studies, the broad methodology could be useful in many other similar situations. Depending on available data, it may be possible to incorporate into models more of the factors detailed above. Doing so will be especially important if the landscape is becoming more
fragmented following disturbance, such that area and edge effects (amongst others) might be reducing biodiversity more than expected in our study. By extending the methods of previous biodiversity models that investigate the impact of land use change on biodiversity (e.g., *Newbold et al., 2015*), the site-level results can be useful to a single decision-maker for a smaller-scale project, especially in relation to planning outcomes. Estimating the long-term impact that disturbance and renovations might have on biodiversity prior to any undertaking can be valuable, especially when results can directly feed into plans and actions to prevent or offset declines in biodiversity. Although other models exist that assess the potential impact on biodiversity of habitat change and loss (e.g., *DEFRA, 2012*), meta-analytical methods, such as these, provide empirical and transparent results.

The redevelopment of the Natural History Museum's grounds provides the opportunity to monitor aspects of biodiversity recovery within an urban environment that have previously been little studied, whilst validating the results found in this study. Establishing long-term ecological sampling within each of the grounds' habitat types would allow more detailed assessment of the recovery of the disturbed habitats as well as the colonisation of the newly created habitats. Standardising the sampling, in conjunction with other projects, would also allow the further comparison of the results with other areas within London (e.g., *Smith, Chapman & Eggleton, 2006*) or the UK more broadly (e.g., BUGS2 project: *Loram et al., 2007*). Long-term regular sampling would also provide the opportunity for other hypotheses to be rigorously tested, for example, whether reduced connectivity in urban areas increases genetic differentiation between populations (*Johnson, Thompson & Saini, 2015*). Of course, with the grounds being an integral part of the Natural History Museum they provide unique opportunities for the monitoring to be undertaken not only by the taxon experts on museum staff but also by members of the public (*Silvertown, 2009*; *Roy et al., 2012*), not only reducing costs but also increasing public engagement in and participation in science and awareness of the new grounds and urban biodiversity in general.

Urban green areas face many threats worldwide. Without robust methods for estimating the consequences for biodiversity, planning decisions run the risk of being uninformed or misinformed. We show how a global modelling approach can be downscaled to inform local decision-makers about the likely impact of habitat change on species richness and species density. In the case of the Natural History Museum, London, the proposed changes to the grounds are predicted to result in a net gain of biodiversity, due to increases in the number and areas of habitat types. The size of the gain depends on the assumptions made about the relationship between within-sample species diversity and habitat area—a relationship that has hitherto largely been ignored in global models. The grounds redevelopment program provides an opportunity for systematic ecological surveys to quantify the effects of habitat creation, expansion, reduction and disturbance in the future, adding useful knowledge about this culturally important urban green space at the same time as allowing improvement of biodiversity models to support planning decisions.

## ACKNOWLEDGEMENTS

We thank John Tweddle and Chris Raper for species names; Mike Sadka for creating extracts from the WLG database; Peter Wilder of Wilder Associates for Fig. 1 and calculating the habitat areas from the planning application; and John Halley for discussions of species–area relationships. We are very grateful to Jonathan Sadler and Seth Magle for extremely helpful review comments. PREDICTS is endorsed by GEO-BON. This is a contribution from the Imperial College Grand Challenges in Ecosystems and the Environment Initiative.

### Funding

This work was funded by the Natural History Museum in relation to their renovation of their grounds. Helen R.P. Phillips was supported by a Hans Rausing Scholarship. Andy Purvis was supported by NERC (grant NE/J011193/2). The funders had no role in study design, data collection and analysis, decision to publish, or preparation of the manuscript.

### Grant Disclosures

The following grant information was disclosed by the authors:
Natural History Museum.
Hans Rausing Scholarship.
NERC: NE/J011193/2.

### Competing Interests

All authors were affiliated with the Natural History Museum, London, whilst undertaking this study. Helen R.P. Phillips and Andy Purvis had no involvement in either the conception of the redevelopment proposal or the opposition to it. Prior to the start of the study, it was agreed that results would be published irrespective of the findings.

### Author Contributions

- Helen R.P. Phillips conceived and designed the experiments, analyzed the data, contributed reagents/materials/analysis tools, wrote the paper, prepared figures and/or tables, reviewed drafts of the paper.
- Sandra Knapp and Andy Purvis conceived and designed the experiments, contributed reagents/materials/analysis tools, reviewed drafts of the paper.

### Data Availability

Github:
https://github.com/helenphillips/GroundsRenovation.
Natural History Museum Data Portal:
http://data.nhm.ac.uk/dataset/grounds-metaanalysis-data.

## Supplemental Information

Supplemental information for this article can be found online at http://dx.doi.org/10.7717/peerj.3914#supplemental-information.

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
