# Peer review of "Estimating the potential biodiversity impact of redeveloping small urban spaces: the Natural History Museum’s grounds"

_PeerJ, doi:10.7717/peerj.3914_

## Round 0.1 · original submission · Minor Revisions

This is a well written and technically correct analysis of the biodiversity impacts of redevelopment of the NHM site. The model, as pointed out by one of the reviewers, is rather basic as it does not take into account well-known factors such as edge effect or connectivity; this needs to be acknowledged in the paper.

Also the authors report there are no conflict of interests. Given the site to be redeveloped is the NHM, the NHM funded the study and at least one author is a paid member of staff while the other I presume is some sort of associate, there is clearly the potential for a conflict of interest. This needs to be acknowledged in the conflicts section. That said, the paper is well written and does not over stretch the findings.

·

Basic reporting

Clear, unambiguous, professional English language used throughout.

This is an exemplary example of a clearly written, structured and produced piece of work. There are very few typographical errors, errors of omission or of expression/fact. I have annotated comments onto the PDF (attached to this review).

Intro & background to show context. Literature well referenced & relevant.

The introduction has a decent review of the literature but there are a couple of points that require some thought. Firstly, I am not persuaded that by the ‘neglected status’ of urban ecology (p.1, line 44ff). There has been a great deal of recent activity in this field. Perhaps rephrase to say just that. I think the lengthy diversion into the history of the grounds redesign at the NHM has too much detail; it might better sit in the methods of supplementary materials. It needs slimming down to about a third picking up just the salient points. The key point of the paper is that it introduces a means of assessing the impact of site-based developments, which is an area where little robust work exists (aside of classic impact assessments). This is the section that requires emphasising.

Structure conforms to PeerJ standard, discipline norm, or improved for clarity.

This is fine.


Figures are relevant, high quality, well labelled & described.

The Figures are nicely produced but would benefit from some modification:
Fig 1: rotate by 90 degrees
Fig. 2: needs scale and north arrow. Not all data sources are listed (in comparison to Github sources).
Fig. 3: Maybe use a 2nd Y-axis to improve clarity and remove the grey text as you describe what it means in the figure caption.
Fig. 5: A bit small to read easily on the screen - increase figure and font size. It is clear on retina-type displays but not everyone will have access to one of those.

Raw data supplied (See PeerJ policy).

Raw data, R code and latex all available on Github @: https://github.com/helenphillips/GroundsRenovation. The code indicates possible sources of problems but looks fine. I have not tried to emulate the analyses.

Experimental design

The design is clearly outlined but there are a few areas where the paper would benefit from more methodological detail and support/justification (all minor stuff):
1. unpack what is meant by the dose-response modelling framework - p.4, line 138
2. I understand why going for a Species|Area-type model is sensible given what you are trying to achieve but lots of variables will impact species richness/density (e.g habitat structure, site context ie. situated in areas with lots of similar habitat or isolate, disturbance histories, time and the like) You allude to this at the end of the introduction but I think you need a develop this a little more and revisit to more directly as a critique in the discussion. I was a little puzzled by line 157 where you mention time limitations; what were they? I cannot see that in the discussion either.
3. Data collation: You need to document what data was extracted using ImageJ.
4. GLMM is a sensible approach but it's a small dataset for inference outside the realms of the study. Some justification of that is needed.
5. Your assumptions are sound given the data but why did you carry out sensitivity analysis on only one of the three?
6. Why were both the density and richness models simplified by removing the additive effects? Elsewhere you mention data paucity as a reason why interactions were not undertaken. Is that the reason?

Validity of the findings

The authors have used an appropriate and robust approach to the analysis and they are careful not overextend the reach of their findings. I like the central idea of this paper as it allows an evaluation of outcomes of site developments thus chimes well with UK planning policy which is essentially site-based at the point of decision. The analytical model / approach I believe to be robust enough to be used elsewhere but it may need nuancing for each specific location. Some guidance on what form this might take in the discussion would be helpful. I think also that a fuller review of the issues surrounded a focus on area is needed as a critique in the discussion.

·

Basic reporting

57-109: This is far too much detail for a study of this type, strongly recommend cutting this down significantly. You are trying to argue that your study has value beyond being a case study, this much detail on the history and specific challenges of your site does not help make this argument.

Experimental design

Line 190: I think this is a central limiting factor of your research. You used data from urban areas “when possible”, but we cannot tell when this was, and a huge wealth of literature tells us that relationships between biodiversity and habitat in urban areas are altogether different than in natural landscapes. This point needs to be made, and strongly. This is a major assumption that casts some doubt on your findings.

Line 227: When scaling in this matter, you completely ignore edge effects, unless I am missing some part of your methods. Given how small your habitat patches are, it seems these effects may be extremely important. And generally, since you are using values from the literature that are collected across much, much wider scales, I don’t think the relationships are likely to scale down to the level of the patches you’re working on.

Validity of the findings

The findings are valid, I think, but the inference space is extraordinarily narrow. You have applied a simulation modelling exercise to a small urban patch using non-urban data, and you haven’t verified it with any field sampling whatsoever. Is this a promising approach for other systems? It might be, though I have some grave concerns about edge effects and scaling as I describe previously. But to test that you’d need to validate it with field data, which this study fails to do.

So what we have in the end is a simulation study, one that tries to use possibly unrelated data to predict biodiversity outcomes for land management on an extremely small scale, and finds those changes to be minimal for total species richness. It’s intuitively obvious that by adding new habitats, and high quality habitats, we would expect these values to go up. Of course, even species richness isn’t usually what we’re interested in, compared to things like diversity of rare species. The method has promise, but I’d like to see it explored over a much wider area, and with some field data to back it up.

At any rate, I think you need to discuss these limitations more fully in the text—you are way out on a speculative limb, and I think the text needs to work harder to acknowledge that.

Additional comments

Thank you for the opportunity to review this manuscript. It is very well written, and in general I applaud studies that investigate the relative value of small patches of urban habitat, which are often much more valuable than people realize. Technically the paper is sound, nothing was done that is misleading or that reaches erroneous conclusions. So for the purposes of this journal I suspect it is publishable.

Unfortunately, I found the potential benefits of the paper for management or conservation lacking. This is a fairly simplistic modelling exercise, applied only to a single case site, that makes a variety of unrealistic assumptions to try to make a relatively simple final observation about how wildlife species might use habitat. I appreciate what the authors are trying to do, but I think the study needs to do at least one or two of the following to be of much use to anyone:

1. Apply their method to more sites
2. Validate their predictions with field data
3. Create a more complex model that incorporates things like edge effects, connectivity, and the like.

I do realize that potential impact is not a determining factor for publication in this journal, so I bring this up largely out of completeness, and in the hopes that the authors would like their work to have a wider impact. I hope this review is useful to the authors.

---

## Round 0.2 · accepted · Accept

Thank you for taking into consideration the reviewers comments. I'm happy with the corrections made and look forward to seeing it in press.